# Burnout, Positivity and Passion in Young Mexican Athletes: The Mediating Effect of Social Support

**DOI:** 10.3390/ijerph18041757

**Published:** 2021-02-11

**Authors:** Julio Román Martínez-Alvarado, Luis Horacio Aguiar Palacios, Yolanda Viridiana Chávez-Flores, Rosendo Berengüí, Ahmed Ali Asadi-González, Ana Gabriela Magallanes Rodríguez

**Affiliations:** 1Faculty of Health Sciences, Autonomous University of Baja California (UABC), 22260 Tijuana, Mexico; aguiar.luis@uabc.edu.mx (L.H.A.P.); viridiana.chavez@uabc.edu.mx (Y.V.C.-F.); gabriela.magallanes@uabc.edu.mx (A.G.M.R.); 2Faculty of Social Sciences and Communication, Catholic University of Murcia (UCAM), 30107 Guadalupe, Spain; 3Faculty of Medicine and Psychology, Autonomous University of Baja California (UABC), 22390 Tijuana, Mexico; ahmedali@uabc.edu.mx

**Keywords:** burnout, passion, positivity, social support, athletes

## Abstract

The burnout syndrome is a negative experience for athlete development and it has been demonstrated that it gets worse when a sport is practiced in an obsessive way. Interventions with a positive view towards sports could be a protective factor to boost the athlete’s wellbeing. The aim of the present study was to analyse the mediator effect from social support, the relationship between the burnout, positivity and passion in young Mexican athletes. The sample was composed by 452 Mexican athletes, males and females (women 45%), from 12 to 18 years of age (*M* = 16.29, *SD* = 1.66). Participants answered the Athlete Burnout Questionnaire, The Scale of the Social Support Perceived by Athletes, the Passion Scale and the Positivity Scale. The results of structural equation modeling showed a good adjustment model (χ^2^ = 889.213; *df* = 274; χ^2^/*df* = 3.245; *p* ˂ 0.01; CFI = 0.93; TLI = 0.91; IFI = 0.94; NFI = 0.91; RMSEA = 0.07). The harmonious passion presented direct and indirect effects on the burnout, being the perceived social support the mediator variable of the indirect effect. The positivity resulted positive predictor from the social support (β = 0.714, *p* ˂ 0.001) and social support predicted the burnout (β = −0.270, *p* ˂ 0.005). The obsessive passion had a direct effect over burnout (β = 0.627, *p* ˂ 0.001). Developing negative commitments to sports could be an indicator of a greater risk of experiencing individual conflicts that lead to sports burnout.

## 1. Introduction

To achieve success in sports, the athlete is exposed to undergo a very demanding training regime since a very young age [1]. Participating in very intensive training and competitions the entire year becomes part of the athlete’s development process to acquire the skills required for high performance [2]. The most talented athletes consider that improvement in performance is a central element in sports [3] and that leads them to increase their effort, even if sports activities become a risk to their own health or wellbeing [4]. One of the negative results from this sports demand is the burnout syndrome [5].

As a result of 50 years of investigation dedicated to the development of burnout in the sport context, researchers have achieved a consensus regarding the definition of burnout [6], which has derived in a measurement of burnout for general use: the Athlete Burnout Questionnaire [7]. In terms of symptomatology, it established burnout as a tridimensional cognitive-affective syndrome characterized by the physical and mental exhaustion, a reduced sense of achievement and devaluation of the sports practice [6]. This definition has its similarities with the Maslach and Jackson definition [8], with a difference in the depersonalization dimension, which in the sports context it is understood as attitudes of devaluation of the sports practice.

Definitively, burnout is a negative experience for the athlete which is composed by elements of physiological, emotional, cognitive, and attitudinal nature [9]. The lack of social support or the little perception of support has been identified as one of the strongest factors that trigger burnout syndrome in athletes [10,11,12,13].

### 1.1. Passion in Athletes

Vallerand and Miquelon [14] defined passion as a strong connection between an individual’s affinity for a sport, along with the amount of time and energy dedicated to it. From this perspective, passion can be understood as the source of energy that motivates the athlete to maintain the commitment and perseverance in any sport [15]. The dualistic model of passion [16,17,18] distinguishes two types of passion: harmonious passion (HP) and obsessive passion (OP), depending on how they are integrated into the individual’s identity.

HP is the result of an autonomous internalization by which the athlete practices sports for joy and pleasure, due to the inspiration caused by the sport itself, more than to meet the expectations that the team or the coach could have on him/her [19]. This passion becomes evident when the person is involved voluntarily, without any pressure, and autonomously in the sports activity. Therefore, the athletes that have HP in a particular sport tend to have a better administration of the time they spend on it without feeling a conflict between the sport and other daily activities [18,20].

In contrast, the OP is associated with a controlled internalization, in which the individual is forced to practice sports searching for social acceptance or better self-esteem experimenting an interiorized pressure in the process [21]. This passion emerges in the absence of satisfaction of the intrinsic needs of the individual that can result in the internalization of intrapersonal or interpersonal pressure, or even both [22,23]. This passion can arise when the activity is rigidly internalized and when the participation is mandatory, which is, at the same time, related with the negative affection [24] and the burnout symptoms.

Unlike the HP, an athlete who experiences OP cannot live without this sport practice and becomes emotionally dependent of it in order to achieve the social acceptance and to search for a personal identity [25]. An obsessively passionate athlete overvalues the importance of the implications of the sports activity as a source to boost self-esteem and a way to escape from problems, making it even harder to stop a pleasant activity [26].

According to findings by Vallerand [17] regarding the sport performance and motivation, it has been shown that negative affect predicts athlete’s burnout [27]. Burnout can lead to several problems, including decrease in performance, feeling frustrated, unmotivated, exhausted, and sometimes depressed [28].

### 1.2. Positivity

Part of the work from researchers in sports psychology is to identify the individual variables that could have an impact on the wellbeing or could conduct to the psychological discomfort (e.g., the burnout syndrome). Some authors have identified the psychological construct of positivity as a long lasting personality characteristic that makes reference to the tendency of the self-evaluation which at the same time predisposes the people to consider life as something valuable, addressing their lives with a positive attitude [29,30]. Positivity can be understood as an evaluation manner, perception and construction that affect in a general manner the way in which the individual is influenced to determined actions and experiences [31]. From this perspective, the concept of positivity could be understood as a protective factor when facing the burnout experience.

Positivity also refers to a tendency of the individual to maintain a positive vision of himself, which could be strengthen by a proper intervention of oriented actions to maintain the wellbeing of people. As a basic disposition characterized by a positive orientation towards the life experience, positivity probably provides resources to help the people face life challenges and, as a last resort, to protect their mental health regardless of the adversities [29], like when the athlete face the negative experience of burnout. Positivity is conceptualized as a positive cognitive orientation to himself, life itself and the individual future. An orientation that emanates positivity could be consider as a good determining factor of the subjective wellbeing that the person has [32]. Furthermore, we should not see this concept as something that is exclusive of the future perception of some event that is to come in life, but as a concept that also considers the self-reference system that occurs before their own human experiences.

### 1.3. Social Support

Social support involves “an exchange of resources between at least two individuals perceived by the provider or recipient to be intended to enhance the well-being of the recipient” [33]. Despite the many definitions, theoretical models and assessment methods proposed, there is agreement in sport that social support can be understood in terms of three main sub-constructs [34,35]. Thus, social support is a form of social integration that refers to the number and types of social ties and relationships within an athlete’s social support network. Also, we can attend to social support as perceived support that derives from a subjective judgement that support is available and can be accessed if needed. Finally, there is support received which refers to the specific actions of help and support provided by individuals within an athlete’s social support network [34,35].

Traditionally, four specific dimensions or forms of social support behavior are accepted: emotional, esteem, informational and tangible [36], and in youth sport the main providers are usually coaches, family and peers [37].

Different studies affirm that social support is positively associated with variables such as well-being, self-confidence, positive appraisal of stress, adaptive responses to injury and sports performance [38,39,40,41,42,43,44,45]. But on the other hand, reduced social support is also associated with negative consequences for the athlete, such as drop-out [37,46] and burnout. It has even been argued that lack of social support is as bad for health as physical inactivity [47]. In the case of burnout, high social support is associated with a lower risk of burnout and protects athletes from its negative impact [41,48,49,50,51].

In relation to the previous concepts, we can consider that people with high levels of positivity generally appreciate social support when they get it [52]. On the opposite, people with negativity have negative cognitive schemes, showing maladjusted conducts which, combined with a low perception of social support, could make the athlete more prone of having a negative experience such as burnout. The analysis of this type of thinking in young athletes, might help to understand the different protective factors that are helpful in order to avoid the burnout syndrome. Due to the above, the objective of this work is to analyse the mediator effect of the social support, in relation with the positivity, the passion and the burnout in young Mexican athletes.

It can be observed in Figure 1 that we tested the following hypotheses: (a) H_1_: the positivity has a direct negative effect on burnout; (b) H_2_: the passion harmonious has a direct negative effect and the passion obsessive a direct positive effect on burnout; (c) H_3_: the effect of positivity, harmonious passion and obsessive passion on burnout is mediated by the perceived social support.

## 2. Materials and Methods

### 2.1. Study Design

This study followed a non-experimental design, where no manipulations of variables were made, only observations. This was a quantitative empirical study carried out through descriptive methods using surveys.

### 2.2. Participants

The sample was selected using a non-probabilistic incidental method. In this study, 452 Mexican athletes of both genders participated (women 45%), with ages between 12 and 18 years (*M* = 16.29, *SD* = 1.66) and have an average of 3 years (*SD* = 1.77) affiliated in a sports team (football 42.5%, soccer 31.4%, volleyball 13.5%, basketball 9.7% and softball 2.7%). These athletes reported an average of 3.57 years (*SD* = 2.44) practicing their sport, to train 3.54 days a week (*SD* = 1.47), mentioning a duration of 2.35 h (*SD* = 0.79) in each training session.

### 2.3. Procedure

The present study was carried out in accordance with the Declaration of Helsinki. Following approval from the local university ethics committee, data collection was conducted between August to December of 2019, with previous authorization of the sports clubs and the athlete’s parents. The application of the instruments was carried out inside the locker room, under the supervision of the main researcher, collectively and self-administrated. The confidentiality of the answers and the willingness of the participants was emphasized. Written consent was obtained from all participants after having explained to them the purpose of the study (a guardian provided written consent in the case of minors).

### 2.4. Instruments

Athlete burnout. In order to measure burnout, the Mexican version of Athlete Burnout Questionnaire was utilized [53]. The original version of ABQ was invented by Raedeke and Smith [7] and it consists of 15 items where the athlete is asked to indicate how frequently he/she presents the symptoms, recording their answers through a Likert scale which oscillates between 1 (not frequently) and 5 (very frequently). Different studies have confirm the reliability and validity of the ABQ [53,54]. The internal consistency reliability of the Mexican version of the ABQ by Cronbach’s was 0.91 for the Physical and emotional exhaustion subscale (PEE), 0.85 for the Devaluation of sports practice subscale (DSP) and 0.74 for the Reduced sense of achievement subscale (RSA). Adequate coefficients of internal consistency were obtained in this study: Cronbach’s alpha = 0.93 (PEE), 0.90 (DSP) and 0.82 (RSA).

Social Support. To evaluate the perceived social support, The Scale of Perceived Social Support by athletes of Cresswell and Eklund was used [55], which consists of five items organized in one single factor, utilizing the Likert scale of five alternatives. Several studies have found appropriate psychometric characteristics [10,56]. Regarding internal consistency, previous studies have shown a Cronbach’s alpha that range from 0.86 to 0.88. In this study, the reliability was satisfactory (Cronbach’s alpha = 0.75).

Passion. In order to measure the passion, the Passion Scale was used [18], which consists in two scales of six items that evaluate the harmonious passion (HP) and the obsessive passion (OP) and four more items to determine the level of passion, using the Likert scale of seven points. Several studies have confirmed the psychometric properties of the scale [57,58]. Cronbach’s alpha ranged from 0.75 to 0.91 in previous studies. Adequate coefficients of internal consistency were obtained in this study: Cronbach’s alpha = 0.92 (HP) and 0.92 (OP).

Positivity. Positivity was evaluated with the positivity Scale (P-scale) of Caprara et al. [59], which is composed by eight items, organized in one factor, that evaluate the positive opinion of the people regarding the self, life, the future and trust in other people. The P-scale, uses a Likert Scale of five points that oscillates between 1 (completely disagree) and 5 (completely in agreement), with a punctuation range between 8 and 40. The highest punctuation refers to a mayor positivity. Several studies have confirmed the adequate psychometric properties of the scale [32,60,61]. The internal consistency of the P-scale by Cronbach’s alpha ranged from 0.75 to 0.89 in previous studies. In this study, Cronbach’s alpha value was 0.87.

### 2.5. Statistical Analysis

The descriptive statistical and normality tests were analysed (mean, standard deviation, skewness and kurtosis). Pearson’s correlation coefficient was calculated between each variable. For the reliability analysis regarding the internal consistency, the Alpha coefficient rate was utilised, using the SPSS 22.0 software (IBM, Armonk, NY, USA).

Convergent validity can be evaluated by examining the factor loadings and Average Variance Extracted (AVE). In general, AVE must be greater than 0.5, indicating that the measurement questions can better reflect the characteristics of each research variable in the model [62]. The strength of the model’s interpretation of the latent variable is the R^2^ value. The larger the R^2^ value, the stronger the model’s interpretation of each latent variable.

Structural Equation Modeling (SEM) was utilized to find the best-fitting model performed with the Maximum likelihood estimation using AMOS 22.0 (IBM, Armonk, NY, USA). To prove the goodness of fit of the model we have implemented the following adjusted indexes: Chi-square statistic (χ^2^), Chi-square divided in degrees of freedom (χ^2^/*df*), the Comparative Fit Index (CFI), the Tucker-Lewis Index (TLI), the Incremental Fit Index (IFI), and the Normed Fit Index (NFI) and The Root Mean Square Error of Approximation (RMSEA).

In order to accept or reject a model, it is advisable to analyze several of the said indexes, making inappropriate a decision based only in one of them. A statistic χ^2^/*df* lower than 5.0 indicates a good adjust of the model. The IFI indicates improvements in the model adjustability on degrees of freedom in comparison with the base line of the independent model. Scores equal to or higher than 0.90 are considered acceptable. The CFI is used to contrast theoretical models using samples of over 100 subjects. This index gathers scores between 0 and 1, recommending scores equal or superior to 0.90 for a good adjustment or superior to 0.95 for an excellent model adjustment [63]. The TLI is an index that considers the degrees of freedom of the proposed model and the null model. Scores equal or superior to 0.90 indicate a good model adjustment. The NFI compares the proposed model and the null model considering a value acceptable if it is higher than 0.90. The RMSEA confirms the unbalanced grade of the covariance matrixes in the theoretical and empirical model. Scores between 0.05 and 0.10 are considered acceptable [64].

## 3. Results

### 3.1. Descriptive Statistics, Internal Reliabilities and Correlations

As we can see on Table 1, the athletes reported low levels in the three factors of burnout, as well as in the total score, obtaining values between 1.88 (*SD* = 1.07) and 1.96 (*SD* = 0.92). On the other hand, moderate high levels in social support were reported (*M* = 4.17, *SD* = 0.80) and positivity (*M* = 4.34, *SD* = 0.79). Regarding the types of passion, average values of harmonious passion were found (*M* = 4.22, *SD* = 0.89) and moderate levels of obsessive passion (*M* = 2.46, *SD* = 1.27).

Regarding the statistical normality, we found scores between 0.17 and 1.50 in skewness and kurtosis respectively. According to the criteria of univariate normality [64], in order to achieve this normality, skewness should be shown under the absolute score of 2 and kurtosis under the absolute score of 7. Considering these results as reference, we can continue with the factorial and structural equations analysis.

According with the criteria established by [65], values in the Cronbach’s Alpha rates equally or superior to 0.70, are considered acceptable and as they get the closest to score 1, they have better reliability. As for the internal consistency reliability, Cronbach’s alpha coefficients indicated superior scores of 0.70 in every case, oscillating between 0.75 and 0.96. Based upon the foregoing, the results revealed that the measuring associated to the instruments utilized, present a good internal consistency.

According with the result obtained in the Pearson correlation analysis, positive and significant correlations were found between the burnout factors and the burnout total score, being the highest correlation the one between the devaluation of sport practice and the total burnout (r = 0.96, *p* ˂ 0.01) and considering these correlations within expectations. On the other hand, negative and significant correlations between the three factors of burnout—positivity, social support and harmonious passion—were found, being the most important correlation the one between the reduced sensation of accomplishment and harmonious passion (r = −0.60, *p* ˂ 0.01). Another expected correlation was between the obsessive passion and the harmonious passion, finding a negative correlation (r = −0.24, *p* ˂ 0.01). Also positive and significant correlations between the burnout factors and the obsessive passion were found, being the most important correlation the one between the obsessive passion and the total burnout (r = 0.66, *p* ˂ 0.01). Positive and significant correlations between the social support, harmonious obsessive and positivity were found, being the most important correlation the one between the social support and positivity (r = 0.63, *p* ˂ 0.01). Along this same line, social support was negatively correlated with obsessive passion (r = 0.29, *p* ˂ 0.01). Finally, we found positive correlations between positivity and harmonious passion (r = 0.59, *p* ˂ 0.01) and a negative correlation between positivity and obsessive passion (r = −0.42, *p* ˂ 0.01).

### 3.2. Convergent Validity

Convergent validity is generally assessed by the loadings of all the items and Average Extracted Variance (AVE). As shown in Table 2, the factor loadings (λ) of almost all the items are above the threshold of 0.70, indicating that the observed variables have high convergent validity. The item 6 of the *P*-scale presented a factorial weight lower than 0.30. Meanwhile, the item 3 of the social support scale presented a negative value. Therefore, both items were excluded from subsequent analyzes. In general, average variance extracted (AVE) is greater than 0.50, which denotes that the latent variables have a convergence ability that is quite ideal [42], except in social support. Table 2 provides the values of δ (1-λ^2^), and percentages of variance explained for each item (R^2^).

### 3.3. Structural Equation Modeling (SEM)

A confirmatory factor analysis (CFA) was conducted to empirically test the measurement model. Model fit was evaluated using the maximum likelihood (ML) method. The specified SEM model is shown in Figure 2, which was comprised of three exogenous variables (positivity, harmonious passion and obsessive passion), one mediator variable (social support) and one endogenous variable (burnout). The adjusted indexes from the observed model showed an appropriate adjust, which allows us to support the initial model (χ^2^ = 889.213; df = 274; χ^2^/df = 3.245; *p* ˂ 0.01; CFI = 0.93; TLI = 0.91; IFI = 0.94; NFI = 0.91; RMSEA = 0.07). The values of RMSEA oscillated between 0.06 and 0.08.

The aforementioned model showed the positivity as a positive predictor from the social support (β = 0.714, *p* ˂ 0.001), the same way the harmonious passion resulted in positive predictor from the social support (β = 0.380, *p* ˂ 0.001) and the obsessive passion as a negative predictor but not significant (β = −0.052, *p* ˂ 0.001) from social support. Finally, the social support predicted the burnout (β = −0.270, *p* ˂ 0.005).

The model explained 50% of the variance of burnout (R^2^ = 0.503). In addition, positivity explained 49% of the variance of social support (R^2^ = 0.490) and harmonious passion 10% of the variance of social support (R^2^ = 0.103). The obsessive passion had an direct effect over burnout (β = 0.627, *p* ˂ 0.001), explained 20% of the variance (R^2^ = 0.207) and the harmonious passion a direct effect over burnout (β = −0.352, *p* ˂ 0.001) explained 9% of the variance (R^2^ = 0.094). Finally, the social support explained 7% of the variance over burnout (R^2^ = 0.076).

### 3.4. Assessment Mediator Effect

Regarding the evaluation of the mediator effect from the social support, the causal steps procedure was utilized [66]. The first step of this procedure requires the specification of a model with direct effects. The second step requires to specify a model with indirect effects through social support. The third and last step compare both models including direct effects as well as indirect effects, this procedure allows to evaluate if the effect from the social support is statistically significant. A mediation could be partial or total, in case that the partial model resulted statistically significant even when the total model indicated a better adjustment. As we can see in Table 3, it was evaluated if the mediator effect from social support regarding the burnout was statistically significant in order to the exogenous variables, positivity, harmonious passion and obsessive passion. The indirect effects are considered significant when the trust intervals of 95% (IT 95%) include the zero. The analysis of the indirect effects proved the mediator effect of the social support on the positivity effects and the harmonious passion over the burnout, not being the case with the obsessive passion. The analysis of the direct effects of both types of passions indicated that the harmonious passion presents a greater negative direct effect over the burnout, and the obsessive passion, a direct positive effect.

## 4. Discussion

The main goal of this investigation was to analyse the mediator effect of social support, in the relation between the burnout, positivity and passion in young Mexican athletes. Based on the dual model of the passion [18], the research sought to understand the roles of harmonious passion and obsessive passion. It would be important to determine if these variables have a direct effect over burnout or if this effect is mediated by social support. On the other hand, based in earlier work on positive orientation [59], we wanted to study the positivity factor as an element associated with the unwellness and test its mediation effects through social support. A negative association with the burnout syndrome would be expected. To choose perceived social support as a mediating variable, we used previous studies as reference [11,67,68,69].

According with the main results, the suggested model presented a good adjustment with the exception of the effect of the obsessive passion mediated by the perceived social support, which did not turn out statistically significant. Additionally, positivity resulted a variable with direct and indirect significant effects over the burnout. The indirect effects were mediated by the perceived social support, which could emphasize the importance of taking care of the social environment of young athletes and its importance in the construction of positive judgments regarding themselves, the life and the future and how these could influence in the maladjusted behavior. The results could go in the direction of supporting the positivity proposal as a personality factor that plays a protective role facing negative experiences as the burnout syndrome in young athletes. The study of personality in the sport field has been a recurring theme over the years. Personality has remained a central element in sport psychology, perhaps due to its predictive power. The contribution of personality to interpersonal relationships has received little attention in the research literature in the sports context, although there is convincing evidence that personality traits are related to important interpersonal outcomes [70]. These results provide empirical evidence that supports the theoretical background of positivity as a predictor of maladjusted behavior or psychological discomfort as depression, negative affections and burnout [71].

Another variable that resulted mediated by the social support was the harmonious passion, which, as with positivity, presented direct and indirect effects statistically regarding the burnout. This result was expected, due to the different studies that have found that the harmonious passion is a significant predictor of burnout [27,72]. Regarding the obsessive passion, it was not measured by the social support but it did obtain a positive direct and significant effect with the burnout. Previous studies have presented reliability difficulties to confirm the indirect effect of obsessive passion over the burnout [67], but a few other have supported the direct effect of obsessive passion over the burnout [27,73]. We recommend to use these data to drive improvement initiatives about burnout prevention programs, based on techniques and exercises to enhance positivity, harmonious passion and social support to reduce the risk of burnout in athletes, improving also their quality of life.

In view of the above, the role of social support must be emphasised. In the context of youth sport, social support is a significant resource that often comes from those closest to the athlete. Sheridan et al. [37] conclude that the combined effects of coach, parents and peer support play an important role in shaping young people’s sport experiences, both from a positive (athletes’ motivation levels, participation in elite sports) and negative (drop-out) perspective. Perceived social support, analysed in this study, has been shown to be beneficial for performance, self-confidence, flow states and burnout [34], is positively associated with improved perceptions of situational control and challenge assessments, and negatively associated with threat assessments [74]. In addition, it can play an important role in protecting athletes from a reduced sense of achievement and protecting them from exhaustion [41,49].

Regarding burnout, different studies have found that high social support is associated with a lower risk of burnout and protects athletes from its damaging impact [41,48,49,50,51]. Physical exhaustion is a key differentiator in influencing burnout in athletes specialising in sport [75]. In addition, social support is related to two basic variables in the origin of the syndrome, stress and motivation. Different models have proposed the benefits of social support on the effects of stress [76], suggesting that social support may moderate its effects, as social support leads to better coping with stress, as coping is reinforced by the supportive actions of others [77]. In addition, social influences are key to the development of adaptive coping responses in young athletes [37]. In terms of motivation, support from coach, parents and peers is associated with higher motivation [78,79]. Also, positive relationships with teammates are associated with self-determined motivation [80], and the motivational climate created by the coach is a predictor of continuation to elite sport [46], with ego-involved climate being associated with a higher risk of burnout [81].

### 4.1. Limitations and Future Research

An important limitation is the age and the performance level of the athletes. Therefore, it is recommended to consider adult population with cognitive characteristics different from young and adolescent athletes in future research. On the other hand, regarding to the positivity personality factor, the high performance adult athletes are in a phase from the vital cycle where their personality development is more stable and defined which could be relevant at the moment of evaluating the positivity influence in the progress of the burnout syndrome. Furthermore, we believe that future investigations could also consider to utilise a multidimensional measure for the perceived social support, which might provide clarity regarding the mediator role of this variable.

We acknowledge a few limitations of the present contribution. Positivity rests on a set of subjective evaluations that are not easily accessible other than through self-reports. Future studies could be implemented to evaluate the usefulness and the suitability of other methods of assessment.

The athletes used in the present study were selected for their availability, so this renders impossible to make statements about the results with greater statistical rigor. Additionally, the heterogeneity of the sample could represent a limitation since the levels of burnout could vary in terms of the type of sport. In the case of this study, sports such as soccer and American football were better represented than softball, basketball, and volleyball. Future research should consider a more equitable distribution regarding the type of sport. Another possibility would be to include individual and team sports in order to analyze the differences in terms of the variables studied.

### 4.2. Practical Implications

Taking into account the results obtained in this study, we consider that they may have relevant implications for sport psychology professionals and researchers in this area. Having analysed the important role of social support, positivity and passion, we believe that the correct intervention is essential for the prevention of burnout and to improve the well-being and quality of life of young athletes.

Coaches and sports institutions should ensure the development of a harmonious passion for sport, so that young athletes engage voluntarily and without pressure in the activity, and can perceive autonomy in practice and that their sport is not an obstacle to other activities in their life [18,19,20]. Furthermore, all significant people around the athlete should strive to foster their positivity as a way to efficiently and profitably live their sporting experience, and with it the challenges and adversities that sport brings with it, and ultimately serve as a means of protecting their mental health [29] and promote subjective well-being [32]. Also, athletes should also be provided with an adequate social support network, which provides protection against stress [77], stimulating adaptive coping responses [37], and even encourage pro-social behavior [82]. To this end, coaches, sports institutions and parents must be provided with the necessary training and education to contribute to the positive experience of sport for young people.

In addition to previous strategies, working on the optimal development of motivation and protecting athlete against stress processes in his or her sport should serve to promote the athlete’s satisfaction, well-being and health, and to prevent burnout and sport dropout.

## 5. Conclusions

The results indicate that positivity has an indirect effect on the burnout syndrome in young Mexican athletes, based in a model where the mediator variable was the perceived social support. At the same time, the perceived social support was a mediator between the effects of the harmonious passion and burnout. Regarding the obsessive passion, no evidence was found about its indirect effect through the social support, but it resulted to have a direct effect over the burnout. Therefore, positivity and harmonious passion are close related with social support.

Thus, studies on variables predicting athlete burnout may greatly contribute in development in personal and social aspects. Hence, research on the relationship between positivity, perceived social support, passion, and burnout in athletes is considered important and necessary in terms of contributing in offering programs in the prevention, and intervention. Clinical applications are also evident due to the role that positivity and passion may play in contrasting a variety of psychological dysfunctions.

This suggests that positivity and harmonious passion should be included when creating effective prevention programs in young athletes to reduce the risk of burnout in order to improve their wellbeing and quality of life. Also, it is important to pay attention to the type of commitment with their sport and personal goals, since the obsessive passion could represent a bigger risk of burnout. Developing negative commitments to sports could be an indicator of a greater risk of experiencing individual conflicts that lead to sports burnout.

## Figures and Tables

**Figure 1 ijerph-18-01757-f001:**
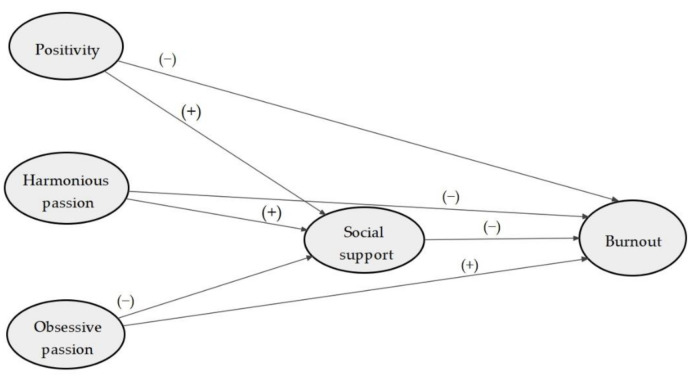
Representation of Structural Model of Positivity, Passion, Social support and Burnout.

**Figure 2 ijerph-18-01757-f002:**
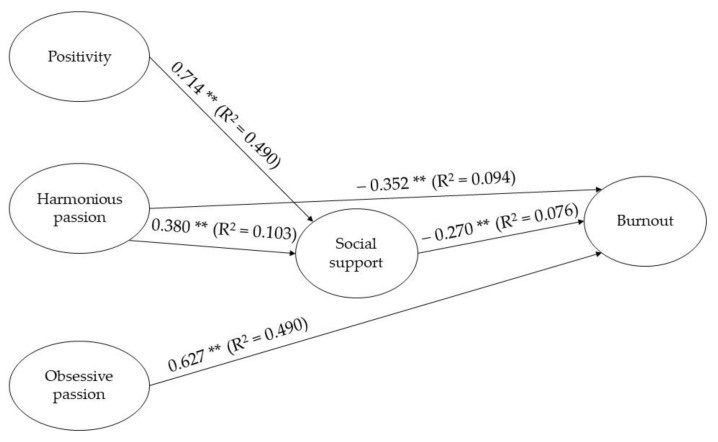
Structural model of positivity, passion, social support and burnout. Note: ** *p* < 0.001; * *p* < 0.005.

**Table 1 ijerph-18-01757-t001:** Descriptive statistics, internal reliabilities and correlations for variables.

Variable	*M*	*SD*	S	K	1	2	3	4	5	6	7	8
1. PEE	1.92	1.03	1.22	0.54	(0.93)							
2. DSP	1.88	1.07	1.11	0.17	0.89 **	(0.90)						
3. RSA	1.96	0.92	0.45	−1.17	0.74 **	0.82 **	(0.82)					
4. Burnout total	1.91	0.95	0.90	−0.29	0.94 **	0.96 **	0.90 **	(0.96)				
5. Social support	4.17	0.80	−1.24	1.50	−0.28 **	−0.34 **	−0.51 **	−0.40 **	(0.75)			
6. HP	4.22	0.89	−1.08	0.67	−0.35 **	−0.42 **	−0.60 **	−0.49 **	0.59 **	(0.92)		
7. OP	2.46	1.27	0.28	−1.26	0.61 **	0.62 **	0.62 **	0.66 **	−0.29 **	−0.24 **	(0.92)	
8. Positivity	4.34	0.79	−1.15	1.01	−0.40 **	−0.43 **	−0.58 **	−0.50 **	0.63 **	0.59 **	−0.42 **	(0.87)

Note: S = Skewness; K = Kurtosis; PEE = Physical and Emotional Exhaustion; DSP = Devaluation of Sport Practice; RSA = Reduced Sense of Achievement; HP = Harmonious passion; OP = Obsessive passion. ** *p* < 0.01.

**Table 2 ijerph-18-01757-t002:** Estimates for all items of the model and AVE.

Scale	Item	Λ	δ	R^2^	AVE
Positivity	POS8	0.70	0.51	0.49	0.60
	POS7	0.77	0.41	0.59	
	POS5	0.81	0.34	0.66	
	POS4	0.81	0.34	0.66	
	POS3	0.77	0.41	0.59	
	POS2	0.82	0.37	0.67	
	POS1	0.75	0.44	0.56	
Harmonious Passion	HP10	0.80	0.36	0.64	0.66
	HP8	0.83	0.31	0.69	
	HP6	0.79	0.38	0.62	
	HP5	0.84	0.29	0.71	
	HP3	0.81	0.34	0.66	
	HP1	0.82	0.37	0.67	
Obsessive Passion	OP12	0.87	0.24	0.76	0.67
	OP11	0.66	0.56	0.44	
	OP9	0.78	0.39	0.61	
	OP7	0.88	0.23	0.77	
	OP4	0.87	0.24	0.76	
	OP2	0.81	0.34	0.66	
Social Support	SS5	0.71	0.50	0.50	0.38
	SS4	0.71	0.50	0.50	
	SS2	0.41	0.83	0.17	
	SS1	0.60	0.64	0.36	
PEE	ABQ2	0.76	0.42	0.58	0.73
	ABQ4	0.86	0.26	0.74	
	ABQ8	0.88	0.23	0.77	
	ABQ10	0.88	0.23	0.77	
	ABQ12	0.89	0.21	0.79	
DSP	ABQ3	0.83	0.31	0.69	0.71
	ABQ6	0.83	0.31	0.69	
	ABQ9	0.86	0.26	0.74	
	ABQ11	0.86	0.26	0.74	
	ABQ15	0.83	0.31	0.69	
RSA	ABQ1	0.41	0.83	0.17	0.50
	ABQ5	0.81	0.34	0.66	
	ABQ7	0.84	0.30	0.70	
	ABQ13	0.86	0.26	0.74	
	ABQ14	0.50	0.75	0.25	

Note: PEE = Physical and Emotional Exhaustion; DSP = Devaluation of Sport Practice; RSA = Reduced Sense of Achievement.

**Table 3 ijerph-18-01757-t003:** Path analysis for the research model, critical ratios, standardized indirect, direct and total effects.

Parameter	Standardized Indirect Effects	Standardized Direct Effects	Standardized Total Effects	C.R.	*p*
Harmonious passion→Burnout	−0.102	−0.352	−0.454	−3.857	***
Obsessive passion→Burnout	−0.009	0.628	0.619	11.164	***
Positivity→Burnout	−0.191	-	−0.191	-	***
Social support→Burnout	-	−0.267	−0.267	−2.779	*
Harmonious passion→Social support	-	0.381	0.381	6.294	***
Obsessive passion→Social support	-	0.033	0.033	0.720	0.471
Positivity→Social support	-	0.714	0.714	9.064	***

Note: *** *p* < *0*.01; * *p* < *0*.005.

## Data Availability

Not applicable.

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
