# Peer review of "Burnout, Positivity and Passion in Young Mexican Athletes: The Mediating Effect of Social Support"

_ijerph, 2021, doi:10.3390/ijerph18041757_

Round 1

Reviewer 1 Report

In the article the authors raise an important and current problem. The article is written correctly, both for recognition in literature, methodology and discussion of the results. However, there are issues worth modifying: - suggest to extract hypotheses and mark them appropriately, which will allow to refer to it in the results in a direct way. - AVE measure is missing, as is loading values for individual items - practice of implication should be developed

Author Response

Dear reviewer,

Kind regards

Reviewer 2 Report

The study design and data collection are well thought out.  The data analysis seems rushed and at times very unclear which points are being made and exactly where the support lies in what is presented.  I suggest that a more thorough and detailed presentation of the data and various analyses be employed.  I think more tables with more detail and more precise explanation would greatly improve the readability of the manuscript.  I also think a visual depiction of the finished causal model would help the reader understand the findings.  I am also confused as to which indirect effect are statistically significant and which direct effect are statistically significant (there are two columns for these, but only one column for a p value - which is which?).   I would expand on the limitations of the study and speak to the mix of participants and the variety of stressors associated with each - is softball as stressful as soccer, for example?  I would also speak to the level of maturity and coping skills of a twelve year old versus an eighteen year old.  I think the manuscript has potential, but needs some refinement as explained above.  I would also suggest a review for English grammar and syntax.

Author Response

Dear reviewer,

Kind regards.

Reviewer 3 Report

This is, in my opinion, an interesting study examining the relationship between positivity and passion (harmonious and obsessive) and burnout. This study also aimed to verify the mediating role of social support. The paper is concise. That being said, I wish to raise some major concerns, hoping that the authors will consider them as helpful.

1-) Citations.

  • On page 1. “Raedeke” should be replaced with the reference number.
  • On page 7. “Vallerand” should be removed from the text.
  • On page 7. Idem for “Caprara”..

2-) It seems that an extensive editing if English language should be done. I have noticed several phrases that were confusing.

For example (among others):

  • On page 3. “... more prone of having a negative experience as the burnout.” I suggest such as burnout.
  • On page 7. ...”we wanted to study the positivity factor as a protective element associated with the wellbeing and that possibly could mediate their effects through social support.” It is very confusing. I suggest modifying that sentence and many others that are similar to that one.
  • On page 7. ... the perceived social support resulted a mediator from the effects of harmonious passion over the burnout.”
  • On page 7. “... provide mayor clarity...”
  • On page 7. ...no evidence was found about the indirect effect through the social support was found...”
  • On page 7. “... in a future, be part of the structural part of prevention programs...”

3-) On page 3. A reference should be added to support the following statement “People with high levels of positivity, generally appreciate when they get support.”

4-) On page 4. “in previous study” should be added to the following sentence: “As to internal consistency, .....”

5-) On page 7. It is mentioned that the results indicate a mediator effect of positivity over burnout. From my understanding, social support was the only mediator that was tested in this study. I suggest to modify that sentence accordingly.

6-) On page 7. I is mentioned that positivity and harmonious passion are protective variables. Considering the cross-sectional nature of this study, I suggest to be more conservative in the conclusions drawn. Protective implies causality/longitudinal design. It could also imply a moderating effect, which was not tested in the present study. A moderation analysis aims to verify whether a third variable can attenuate or accentuate the effect of an independent variable on a dependent variable.

7-) The results should be explained further in the light of previous studies and also of the theoretical framework used.

8-) The discussion should be further elaborated, especially regarding the limits of the study. For examples:

  • The use of self-report questionnaire should be discussed (potential biases?).
  • The cross-sectional design should be discussed.
  • It is unclear why age and performance level of the athletes are considered important limitations.

9-) Suggestions for future research should be extended. It is unclear why age and performance level of the athletes are considered important limitations.

10-) Practical implications should be discussed more clearly.

11-) Did this study received ethical approval by an Institutional Review Board (IRB), Ethical Committee ?

Author Response

Dear reviewer,

Kind regards.

Reviewer 4 Report

Dear authors,
The manuscript postulated examines an interesting process as the mediating role of social support with regard to the burnout syndrome a population of young mexican athletes. However, the manuscript need introduce several corrections explained in the attached file in order to increase it overall merit.

  • A deepen theoretical analysis of social support is missing in all the manuscript. The vast literature on social support tends to differentiate between different domains of this psychosocial process, that amust be having into consideration when this variable is included as a mediator in a research design.
  • From my point of view, probably, a mediation analysis could be a simplistic and more effective methodological option than SEM.
  • Repprt Alphas of this stuydy in all scales applied in the research.
  • Authors must to explain what is the contribution of this research to improve well-being and quality of life of young athletes. One way to do this is explaining the especific contributions of this investigation to increase the effectivenness of intervention programs aimed to reduce burnout in this popultion. However, this contribution does not appear in the manuscript. At the same time, the discusion is the place to exhibit what add this manuscript to the theory on social support in the population evaluated in this work. Something that also is missing.

Regards,

Author Response

Dear reviewer,

Kind regards.

Round 2

Reviewer 2 Report

The article offers new and important insight into burnout among younger athletes and suggests possible interventions via social support mechanisms to protect the athletes well being.  The study is methodologically sound, the data analysis appropriate and results clearly explained.  I enjoyed reading the study. 

Reviewer 3 Report

I thank the authors for addressing the points raised during the first review.

In this sentence: "According with the criteria established by Nunnally [66]." Nunnally shoulb be remove. 

Reviewer 4 Report

Dear authors,

Thank you for your efforts to improve the overall quality of the manuscript. After to review again the new version of the article, I consider that the problems detected in the firts round of the revision have been corrected.

Regards,